# Diabetes and Pancreatic Cancer—A Dangerous Liaison Relying on Carbonyl Stress

**DOI:** 10.3390/cancers13020313

**Published:** 2021-01-16

**Authors:** Stefano Menini, Carla Iacobini, Martina Vitale, Carlo Pesce, Giuseppe Pugliese

**Affiliations:** 1Department of Clinical and Molecular Medicine, “La Sapienza” University, 00189 Rome, Italy; stefano.menini@uniroma1.it (S.M.); carla.iacobini@gmail.com (C.I.); vitale.martina1987@gmail.com (M.V.); 2Department of Neurosciences, Rehabilitation, Ophtalmology, Genetic and Maternal Infantile Sciences (DINOGMI), Department of Excellence of MIUR, University of Genoa Medical School, 16132 Genoa, Italy; pesce@unige.it

**Keywords:** pancreatic ductal adenocarcinoma, hyperglycemia, obesity, reactive carbonyl species, methylglyoxal, receptor for advanced glycation end-products, carnosine derivatives, yes-associated protein, epithelial growth factor receptor

## Abstract

**Simple Summary:**

Diabetic people have an increased risk of developing several types of cancers, particularly pancreatic cancer. The higher availability of glucose and/or lipids that characterizes diabetes and obesity is responsible for the increased production of highly reactive carbonyl compounds, a condition referred to as “carbonyl stress”. Also known as glycotoxins and lipotoxins, these compounds react quickly and damage various molecules in cells forming final products termed AGEs (advanced glycation end-products). AGEs were shown to markedly accelerate tumor development in an experimental model of pancreatic cancer and AGE inhibition prevented the tumor-promoting effect of diabetes. In humans, carbonyl stress has been associated with the risk of pancreatic cancer and recognized as a possible contributor to other cancers, including breast and colorectal cancer. These findings suggest that carbonyl stress is involved in cancer development and growth and may be the mechanistic link between diabetes and pancreatic cancer, thus representing a potential drug target.

**Abstract:**

Both type 2 (T2DM) and type 1 (T1DM) diabetes mellitus confer an increased risk of pancreatic cancer in humans. The magnitude and temporal trajectory of the risk conferred by the two forms of diabetes are similar, suggesting a common mechanism. Carbonyl stress is a hallmark of hyperglycemia and dyslipidemia, which accompanies T2DM, prediabetes, and obesity. Accumulating evidence demonstrates that diabetes promotes pancreatic ductal adenocarcinoma (PDAC) in experimental models of T2DM, a finding recently confirmed in a T1DM model. The carbonyl stress markers advanced glycation end-products (AGEs), the levels of which are increased in diabetes, were shown to markedly accelerate tumor development in a mouse model of Kras-driven PDAC. Consistently, inhibition of AGE formation by trapping their carbonyl precursors (i.e., reactive carbonyl species, RCS) prevented the PDAC-promoting effect of diabetes. Considering the growing attention on carbonyl stress in the onset and progression of several cancers, including breast, lung and colorectal cancer, this review discusses the mechanisms by which glucose and lipid imbalances induce a status of carbonyl stress, the oncogenic pathways activated by AGEs and their precursors RCS, and the potential use of carbonyl-scavenging agents and AGE inhibitors in PDAC prevention and treatment, particularly in high-risk diabetic individuals.

## 1. Introduction

Pancreatic cancer is a highly fatal malignancy with very poor overall survival rates. Pancreatic ductal adenocarcinoma (PDAC) is by far the most common and most lethal type of pancreatic cancer, representing over 90% of the pancreatic malignancies [1]. According to the latest data from the Surveillance, Epidemiology, and End Results Program, the five-year survival rate for people with PDAC is 9% [2]. Because of the poor survival outcomes, PDAC is the seventh leading cause of global cancer death despite being the 10th most common incident cancer [3] and is projected to become the second leading cause of neoplasia-related deaths in the USA in the next two decades [4], in parallel with the rising prevalence of risk factors such as obesity and diabetes. The main causes of these dismal outcomes and prospects are multifactorial in nature; as no simple early detection methods exists, PDAC patients are predominantly elderly people in overall poor health, and PDAC tumors have the ability to rapidly develop acquired resistance to therapies. What is more, the peculiar ability of PDAC to metastasize early in the disease course limits the effectiveness of surgery and radiation [5]. Therefore, in the absence of effective and efficient diagnostic and therapeutic tools, it is important to focus on prevention by eliminating modifiable risk factors associated with PDAC or, at least, defuse the threat they pose by identifying the molecular mechanisms underlying their PDAC-promoting activity. Among these risk factors, diabetes mellitus has been shown to have a remarkable association with PDAC [6].

Driven by the pandemic of obesity, diabetes is a growing global public health issue contributing to premature mortality, morbidity, and disability [7]. Diabetes prevalence is steadily increasing everywhere, particularly in the world’s middle-income countries [8]. The number of adults living with this metabolic disease increased from 108 million in 1980 to 422 million in 2014 and is predicted to rise to 642 million by 2040 [9] (i.e., over 10% of the global population). According to the American Diabetes Association definition, diabetes mellitus is a group of chronic metabolic diseases characterized by abnormal metabolism of carbohydrates secondary to defects in insulin secretion, action, or both, resulting in high levels of glucose in the blood [10]. Type 2 diabetes mellitus (T2DM) is the most prevalent form of diabetes, accounting for 90–95% of diabetic patients. Often associated with overweight and obesity, hyperglycemia is the result of resistance to insulin action combined with inadequate insulin secretion [11]. Formerly known as juvenile-onset diabetes or insulin-dependent diabetes, Type 1 diabetes mellitus (T1DM) accounts for less than 10% of diabetic patients and is characterized by an absolute deficiency of insulin secretion due to immune-mediated destruction of the insulin-producing β-cells of pancreatic islets [10]. In addition to these etiological distinctions, important metabolic differences exist between T1DM and T2DM, as dyslipidemia and hypertension often pre-date diagnosis and accompany T2DM, but usually not T1DM. In the same way, hyperglycemia, which is the common factor of the two forms of diabetes, can arise a long time before T2DM diagnosis. The Centers for Disease Control and Prevention estimates that in the USA, for every known case of diabetes, there is one undiagnosed case of T2DM and one with prediabetes (i.e., impaired fasting glucose or impaired glucose tolerance) [11]. This observation has important clinical implications since the toxic effects of glucose may induce pathological and functional changes in different body tissues and organs in the absence of overt clinical symptoms [10]. Along with hyperglycemia, its consequences are also shared by the two forms of diabetes, as chronic complications such as cardiovascular and kidney disease, vision loss, and neurological deficits affect both T1DM and T2DM patients in the long run [12]. Finally, both T2DM and T1DM have been increasingly recognized as risk factors for the development of various cancers, including PDAC [6,13,14,15,16].

Following a brief presentation of the complex dual relationship between pancreatic cancer and diabetes, this review will summarize the epidemiological data and experimental evidence on T1DM and T2DM as risk factors for PDAC. In particular, the mechanisms by which glucose and lipid imbalances drive diabetes-associated carbonyl stress, the role of carbonyl stress in cancer, especially PDAC, and the potential use of carbonyl-scavenging agents in PDAC prevention/treatment in high-risk diabetic individuals will be discussed in detail. Though diabetes has been recently identified as possible risk factor for other types of pancreatic cancer, particularly the intraductal papillary mucinous neoplasm of the pancreas [17] and pancreatic neuroendocrine tumors [18], the amount of available data on the relationship between non-PDAC tumors and diabetes is limited. Therefore, this review is restricted to PDAC.

## 2. Relationship between Diabetes and Pancreatic Cancer

The association of diabetes and pancreatic cancer has been observed for almost two centuries [19]. Compared to common cancers, the prevalence of diabetes in PDAC is more than three times [6], and the increased incidence of PDAC in the diabetic population has been observed in several epidemiologic studies [6,20,21,22,23]. However, in over 70% of diabetic patients with PDAC, the diagnosis occurs just before, concurrently, or within 24 months after diagnosis of diabetes [24]. As a result, the highest risk for PDAC is observed within the first two years after diabetes diagnosis (4–7-fold). Then, the risk conferred by diabetes gradually decreases to nearly 2-to-4-fold between the second and fourth year after diagnosis, and to 1.5-to-2-fold thereafter [25,26,27] (Figure 1).

These figures highlight the complexity of the association between PDAC and diabetes. In fact, most epidemiological data point to PDAC as both a cause and a consequence of diabetes [28], thus indicating “dual causality”. While long-standing (>3 years) diabetes has been definitely recognized as a risk factor [20,21,22,25,26,27,29], PDAC is considered a possible cause of hyperglycemia in patients diagnosed with this malignancy within 24–36 months after identification of diabetes [6,13,15,24,30]. In this scenario, the ability to recognize new onset pancreatogenic diabetes as an early manifestation of PDAC (i.e., as distinct from T2DM) would represent a significant development for the oncology community, allowing for the diagnosis of early, potentially resectable tumors.

Unfortunately, there are currently no established diagnostic criteria for differentiating T2DM from diabetes that occurs as an early consequence of PDAC [6,31]. The efforts to find a signature to identify diabetes secondary to pancreatic exocrine disease (i.e., pancreatogenic or Type 3c diabetes) and to distinguish it from T2DM have not yet given the desired results [31,32]. In addition, despite some indications in favor of it, the hypothesis of a diabetogenic effect of early PDAC has not been definitely confirmed. The more convincing evidence in favor of new-onset diabetes as a paraneoplastic phenomenon is that it may resolve following PDAC resection [24,33]. However, it should be noted that diabetes remission after bariatric surgery has been observed also in patients with pancreatic diseases other than cancer and in obese patients, even before significant body weight reduction [31,34]. Therefore, the resolution of hyperglycemia may be related to gastrointestinal anatomic changes associated with the specific surgical procedures [35], more than to cancer removal. Finally, given the existence of a large and growing number of subjects with undiagnosed T2DM or prediabetes (one of each for every diagnosed case of T2DM) [10,11], it cannot be ruled out that PDAC diagnosis may simply unmask pre-existing undiagnosed T2DM or that early undiagnosed PDAC may precipitate overt T2DM in prediabetic subjects. These two eventualities would explain, respectively, the high likelihood of concurrent diagnosis and the high incidence of PDAC in close temporal proximity to diabetes diagnosis. These would also imply that abnormal glucose metabolism (i.e., impaired fasting glucose, impaired glucose tolerance, or even overt diabetes) is actually present long before the onset of PDAC and may be a contributor to its development. Consistent with this assumption and suggestive of a possible causal role of impaired glucose homeostasis, PDAC incidence [36,37] and mortality [38,39,40] have been recently shown to increase with increasing fasting glucose levels, even within the normal range [36,37]. Moreover, in a recent case-control study, Sharma et al. provided evidence that blood sugar levels in PDAC patients were elevated for up to three years prior to PDAC diagnosis [41]. Although even this finding does not definitely clarify whether hyperglycemia is the cause or effect of PDAC, it provides an argument in favor of a screening strategy of individuals with new-onset hyperglycemia for PDAC diagnosis at earlier stages.

These considerations apply to the relationship between PDAC and T2DM, which is by far the most common form of diabetes. Mainly because of its overwhelming prevalence, T2DM has been traditionally thought to be more related to PDAC than T1DM [15,42]. As a result of this, most studies have either been restricted to people with T2DM or have made no distinction between types of diabetes. Actually, more recent epidemiological investigations have shown that T1DM is also a risk factor for PDAC [14,15,16] and other malignancies [14,43], including liver, kidney, and stomach cancers [14]. In 2007, a systematic review of nine studies analyzing pancreatic cancer risk by diabetes subtype reported that the overall relative risk for PDAC in T1DM compared with nondiabetic subjects was 2.0 [16], i.e., the same as T2DM [25,26,27]. More recently, Carstensen and collaborators confirmed that T1DM is associated with a long-term risk for PDAC similar to that of T2DM and is a risk factor for other malignancies previously associated to T2DM [14]. As for T2DM, the risk for both pancreatic and other diabetes-related cancers follow an inverse trend with diabetes duration, as the risk is higher in recently onset T1DM and gradually decreases over time. In addition to indicating that also T1DM has a complex relationship with PDAC and cancer in general, these findings have important mechanistic implications that deserve to be considered to depict a more complete picture of the nature of the association between diabetes and PDAC. Indeed, the observation that T1DM and T2DM confer a similar risk, both in terms of magnitude and temporal trajectory, suggests a common mechanism related to hyperglycemia and a rapid effect of diabetes on the development of PDAC, irrespective of diabetes type.

To make things even more complex, obesity was also proposed to be a possible causal factor of PDAC, as body mass index is also associated with a modest increase in risk [44,45,46,47]. Based on the observation that excessive energy intake, elevated body mass index, and central obesity have been reported to increase the risk of both PDAC [48] and diabetes [49,50], it cannot be excluded that T2DM and PDAC share common pathogenic mechanisms related to insulin resistance/hyperinsulinemia and/or chronic metabolic inflammation (also known as metaflammation). However, it has not been definitely clarified whether obesity per se or obesity-related metabolic conditions, including abnormal glucose metabolism, hyperlipidemia and metaflammation, mediate the association with PDAC [51,52,53,54]. A recent Korean nationwide study demonstrated that a metabolically unhealthy phenotype was associated with an increased risk of pancreatic cancer regardless of body mass index, suggesting that metabolic abnormalities might represent a risk factor for PDAC independently of obesity [55]. Several experimental studies have shown that obesity-related T2DM, either induced by a high-fat diet [56,57,58,59,60,61,62] or generated by a genetically engineered deletion of leptin [63], promotes PDAC through various mechanisms [56,57,58,59,60,61,62,63], including enhancement of aerobic glycolysis in tumor cells [58] and systemic inflammation [60,61]. These rodent models of obesity-related T2DM recapitulate all the key features of human metabolic syndrome, including increased adiposity, insulin resistance, glucose and lipid abnormalities, and metaflammation. Although of great clinical relevance, the studies conducted in these animal models of T2DM do not allow to evaluate the contribution of every single risk factor and cannot help to search for a single unifying mechanism to explain the increased risk of PDAC observed in both T2DM and T1DM.

Finally, another important issue to consider in the relationship between diabetes and PDAC is the effect of diabetic medications. Unfortunately, even in this case, no definitive conclusions have as yet been reached. It is not entirely clear whether, how much, and in which direction diabetes treatments affect the association between diabetes and PDAC [23,25,64,65,66,67]. To give a couple of examples, the protection provided by metformin, the most commonly drug used to treat T2DM, is currently under vigorous discussion, together with its mechanistic implications [67,68,69,70,71,72,73], and even the increased risk of PDAC observed in insulin-treated patients in some case-control studies [27,74] can be misleading [26,72]. In fact, it cannot be ascertained whether this effect is attributable to the mitogenic stimulus of insulin on tumor cells or reflects the severity of diabetes and the difficulty to treat hyperglycemia with oral agents. In general, with reference to the effects of oral diabetes medications, the evidence for specific class effects on PDAC risk is largely inconsistent, as a meta-analysis of data from 15 case-control studies indicated that the reduced risk of pancreatic cancer associated with the use of these agents seems to be related to their glucose lowering effect [67]. In particular, regarding thiazolidinediones, despite the initial excitement over the possible prevention and therapeutic potential of these medications in pancreatic cancer [75], a functional network study [76], and a cohort and nested case-control study among persons with diabetes found an increased risk of pancreatic cancer associated with ever use of pioglitazone [77].

Overall, studies designed to unravel the mechanistic link between diabetes and PDAC are complicated by the fact that T2DM and the metabolic abnormalities clustering with hyperglycemia in the metabolic syndrome might promote PDAC through a variety of factors, including hyperglycemia itself, obesity, dyslipidemia, and insulin resistance/hyperinsulinemia [78,79,80]. The coexistence of several potential risk factors suggests a systematic approach to analyze the effect of each candidate risk factor (i.e., hyperglycemia, dyslipidemia, etc.) and the related molecular mechanisms. On the other hand, it is also worth investigating whether all of these metabolic risk factors share a common molecular mechanism.

## 3. Diabetes, Carbonyl Stress, Advanced Glycation End-Products (AGEs) Formation, and Related Therapeutic Strategies

The term carbonyl stress is used to describe a condition characterized by a generalized increase in the steady-state levels of reactive carbonyl species (RCS). These are unstable carbonyl compounds, especially aldehydes, formed by both oxidative and nonoxidative reactions of carbohydrates and lipids [81] (Figure 2).

In diabetes and obesity-related metabolic disorders, RCS overproduction may derive from both glucose [82,83,84], which is increased in both T1DM and T2DM, and lipids [85,86,87], which are usually increased in obese individuals regardless of the presence of prediabetes or T2DM. For instance, glyoxal, malondialdehyde, and 4-hydroxynonenal, three major RCS and biologically active aldehydes [88,89,90], may be produced in blood and other tissues by oxidative modifications of circulating sugars and lipids (i.e., glyco- and lipoxidation reactions) [85,86]. In addition, intracellular metabolism of excess glucose through the glycolytic pathway leads to increased production of the toxic glycolytic side products 3-deoxyglucosone and methylglyoxal (MGO), two highly reactive dicarbonyls [84,91,92,93]. The higher availability of glucose and/or lipids (i.e., increased substrate availability or substrate stress) is the main mechanism responsible for the increase in carbonyl stress in diabetes and related metabolic disorders, though chronic overload and/or deficiencies in the metabolic pathways involved in detoxification of these toxic compounds may contribute to their build-up in tissues [81].

By reacting with free amino groups and thiol groups, RCS induce physico-chemical modifications of proteins, lipids, and nucleic acids that affect many functions of these biomolecules, including immunogenicity, half-life, enzymatic activity, and ligand binding [7]. Depending on the carbohydrate or lipid nature of the substrate from which RCS origin, the final reaction products are defined as advanced glycation end-products (AGEs) or advanced lipoxidation end-products, respectively. Actually, the distinction based on the chemical nature of the substrate from which originate the RCS has little, if any, pathophysiological relevance, as most of the RCS and their final end-products present similar structural motifs [89,90] or are even identical [85]. For example, the AGE N^ε^-carboxymethyl-lysine (CML), a major AGE epitope recognized in vivo [94], originates form covalent adduction of nucleophilic amino acids by glyoxal, an RCS that derives from both carbohydrates and polyunsaturated fatty acids during glyco- and lipoxidation reactions [85]. In addition, most of these final and stable compounds exert their dangerous effects through the same molecular mechanisms [89,90]. Therefore, from here on we will collectively call AGEs the final products of carbonyl stress.

AGEs accumulate in sera and tissues during the ageing process because of glycolytic and oxidative reactions, reduced activity of the detoxification systems, cigarette smoking, and consumption of high-temperature-processed foods [95,96,97]. The rate of AGE build-up is accelerated in several disease conditions, including obesity, dyslipidemia, atherosclerosis, renal and liver diseases, other chronic inflammatory disorders, and, particularly, diabetes [81,98]. Of note, exogenous AGEs, in particular those derived from the diet, have been claimed to contribute to several disease processes, including cancer in general and, specifically PDAC, as comprehensively reviewed in [99,100] and demonstrated in [101].

Although chemically inert, AGEs elicit cellular responses through binding to the receptor for AGEs (RAGE), a 35 Kilodalton transmembrane receptor of the immunoglobulin super family able to detect a class of ligands through a common structural motif. For this reason, RAGE is often referred to as a pattern recognition receptor [102,103]. Also called AGER, its name obviously derives from its ability to bind AGEs. Therefore, in addition to exert direct effects brought about by RCS reactions, carbonyl stress can induce indirect biological effects through binding of AGEs to receptors of the innate immune system and induction of a chronic inflammatory response [104]. In particular, AGE binding to RAGE activates transcription factors and redox sensitive signaling pathways leading to reactive oxygen species formation, inflammation, fibrosis, autophagy, proliferation, etc. [105,106] (Figure 2). Accordingly, besides representing reliable biomarkers of carbonyl stress and tissue damage [81], AGEs are thought to contribute to the development of several disease conditions, including diabetes-related metabolic disorders, their vascular complications, and cancer. As a whole, carbonyl stress may therefore affect cell and tissue homeostasis through a number of mediators (i.e., RCS, AGE/RAGE axis, ROS), each able to affect the cellular redox status, thereby leading to the activation of the redox-sensitive transcription factor nuclear factor-κB (NF-κB), which regulates hundreds of genes involved in cellular stress responses and survival. For this reason, the prevention of AGE formation by trapping RCS or the removal of the already formed AGEs are considered suitable strategies to prevent carbonyl stress/AGE-related diseases, and novel therapeutic agents endowed with these properties are currently under intensive investigation [107,108,109,110,111,112,113,114,115,116,117,118].

Based on the biochemistry of AGEs and their precursors RCS, an efficient therapeutic strategy against carbonyl stress should be directed at reducing AGE formation by quenching RCS derived from both oxidative and non-oxidative metabolism of excess glucose and lipids. A suitable alternative may be the enhancement of RCS degradation by induction of detoxifying enzymes. Conversely, although tested in preclinical studies with encouraging results [116,117,118], the breakage of pre-existing AGE cross-links or blockade of RAGE signaling cannot prevent structural and functional modifications of biomolecules by RCS [89,114,119]. RCS scavengers include hydrazine derivatives, such as hydralazine, aminoguanidine, and OPB-9195, vitamin B derivatives, such as pyridoxamine, thiamine, and benfotiamine, and amino acid derivatives, such as N-acetyl cysteine, histidyl hydrazide, and carnosine [120]. Some of these agents, namely aminoguanidine [121] and vitamin B derivatives [122], including pyridoxamine [123,124], have been investigated in human trials of diabetic nephropathy with inconclusive results, due to safety concerns and inconsistent efficacy. In case of aminoguanidine, the disappointing results have been attributed to its promiscuous activity and lack of selectivity, the latter due to the cross reactivity with physiological aldehydes, such as pyridoxal [120]. Moreover, though vitamin B6 has shown anti-tumor activities in vitro [125], no study has been conducted so far in cancer, including PDAC.

In order to avoid safety issues, attention has been focused on endogenous compounds with proven RCS scavenging activity. L-carnosine (beta-alanyl-L-histidine) is a naturally occurring dipeptide particularly abundant in the nervous system, skeletal muscle, and kidney [126,127] (Figure 3A).

Although its biochemical role has not yet been elucidated, a growing body of evidence indicates that this endogenous compound acts as a quencher of RCS derived from: (1) Lipoxidation, including malondialdehyde [128], HNE, and acrolein [107]; (2) glucose oxidation (i.e., glyoxal) [110]; and (3) excessive intracellular glucose flux through the glycolytic pathway, including the reactive dicarbonyl MGO generated as an inevitable by-product of glycolysis [93,110,129,130,131]. Because of its ability to quench RCS, inhibit AGE formation, and prevent the activation of pro-oxidant and inflammatory pathways, L-carnosine supplementation has been tested with encouraging results in several disease models in which carbonyl stress is thought to play a central pathogenic role, including diabetes, obesity and related vascular complications [112,114,132,133,134,135]. Interestingly, L-carnosine was also shown to be effective in counteracting glycolysis-dependent tumor growth by quenching MGO [129]. Unfortunately, in humans, L-carnosine has a short half-life due to its rapid inactivation by serum carnosine dipeptidases [132,136,137]. Therefore, the search for carnosinase-resistant carnosine derivatives represents a suitable strategy against carbonyl stress-dependent disease conditions. In particular, diabetes and related metabolic disorders may benefit from treatment with these compounds to abate carbonyl stress resulting from hyperglycemia and dyslipidemia. Among the novels bioavailable compounds, the L-carnosine derivative carnosinol, i.e., (2S)-2-(3-amino propanoylamino)-3-(1H-imidazol-5-yl) propanol (FL-926-16) [110,111,114,138] (Figure 3B) and the enantiomer D-carnosine [112,113,138] (Figure 3C) were shown to be highly effective in attenuating obesity-related metabolic dysfunctions [110,111], and vascular complications of both diabetes [112,114] and dyslipidemia [113].

## 4. Carbonyl Stress in Cancer: A Possible Link between Metabolism and Malignances

Because of their potent cytotoxic activity, RCS were tested in preclinical settings as potential therapeutic agents in cancer. Despite the initial positive results [139,140] the higher toxicity to normal cells than to the cancer cells excluded any potential use of these compounds in human therapy. What is more, as observed for oxidative stress and many other biological processes [141,142,143], more recent investigations have shown that carbonyl stress follows a hormetic dose–response model, as RCS, particularly the dicarbonyl MGO, have a biphasic effect on cancer cell viability. Indeed, the exposure to a physiological range of MGO concentrations, lower than the pharmacological concentrations required to inhibit tumor growth, induces an adaptive beneficial effect on cancer cells, favoring their proliferation [144]. The same biphasic effect was demonstrated for other RCS [145], including the lipoperoxidation product 4-hydroxynonenal [146]. Therefore, more than as a possible new tool in cancer therapy, these findings suggest RCS as possible contributors to cancer onset and progression.

Altered energetic metabolism is a common feature of malignancies. Even in the presence of oxygen, cancer cells tend to favor aerobic glycolysis over the mitochondrial respiration pathway for ATP production. Upon this metabolic rewiring, also known as Warburg effect, tumor tissues have high rates of glucose uptake and utilization [147]. In human PDAC, a positive association between the expression of the glucose transporter 1 (GLUT-1) and the histological grade of dysplastic lesions or tumor size has been observed [148]. This is consistent with the finding that PDAC cells show an increased rate of 18F-fluorodeoxyglucose (18F-FDG) uptake, indicating an accelerated glucose metabolism possibly driven by oncogenic Kristen rat sarcoma viral oncogene homolog (KRAS) [149].

In cancer, a positive relationship exists between the rate of glucose metabolism and that of cell proliferation [150]. Accordingly, inhibiting glycolysis or withdrawing glucose is deleterious to cancer cell proliferation and tumorigenesis in experimental models [151,152]. However, how glycolysis is related to cell proliferation is still not fully understood. A partial explanation is that the Warburg effect allows cancer cells to maintain large pools of glycolytic intermediates to support anabolic metabolism by feeding several biosynthetic pathways that branch from glycolysis [153,154] (Figure 4).

However, in addition to supply nucleic acids, amino acids, and lipids for the synthesis of new cellular constituents, may enhanced glycolytic flux also directly stimulate cell proliferation? Recent advances have provided evidence supporting this hypothesis, as the glycolysis-derived RCS MGO has been demonstrated to be a potent inducer of cell proliferation [129]. This finding opens up new perspectives in the search for a link between diabetes and cancer. In fact, despite the finding from several studies that hyperglycemia is associated with an increased cancer risk, progression, and mortality [155], there is scarce information on the possible mechanistic link between altered systemic glucose metabolism and cancer. Taking into consideration the notion that cell proliferation is stimulated by MGO, hyperglycemia might favor cancer growth by both ensuring unrestricted glucose availability to glycolysis-dependent cancer cells [154] and providing additional fuel for cell proliferation in the form of carbonyl stress [129].

An unavoidable consequence of increased glucose uptake and glycolytic flux is the accumulation of toxic glucose metabolites such as RCS and their irreversible adducts AGEs [93,131,156] (Figure 4). A causative link between the α-oxaldehide MGO and breast cancer aggressiveness has been established [129]. MGO is one of the main by-products of glycolysis with a critical role in the glycation process to form AGEs. MGO and MGO-derived AGEs are inevitably produced from the spontaneous degradation of dihydroxyacetone phosphate and glyceraldehyde 3-phosphate (Figure 4), two triose phosphates, and glycolytic intermediates, the levels of which are increased even in normal cells when exposed to glucose concentrations in the diabetic range [156,157]. Accordingly, the diversion of glycolytic intermediates into the polyol, hexosamine, and diacylglycerol pathways, and accumulation of RCS/AGEs are widely recognized as critical mediators of diabetic complications [157,158]. Noteworthy, Bellahcène’s group recently demonstrated that accumulation of AGEs is a common feature of breast cancer [129,159] and that MGO-mediated glycation promotes breast cancer progression and invasiveness by inducing extracellular matrix remodeling and the activation of signaling pathways promoting survival and migration [160]. In addition, MGO stress was identified as a constant feature of KRAS-mutated colorectal tumors [161], and RCS/AGE accumulation was also found in melanoma tissues [162]. Mechanistically, MGO was demonstrated to affect heat-shock protein 90 (Hsp90) chaperone activity by inducing post-translational modification of several lysine and arginine residues, leading to the formation of the AGEs carboxyethyllysine and argpyrimidine/hydroimidazolone adducts. This resulted in reduced ATPase and binding activity of Hsp90 to large tumor suppressor 1 (LATS1), a key kinase of the Hippo pathway involved in phosphorylation-mediated inactivation of Yes-associated protein (YAP) [129]. Accordingly, cancer cells with high MGO stress showed persistent nuclear localization and activity of YAP, a key downstream target of KRAS signaling [163] and transcriptional co-activator regulating tumor growth and invasion [164]. Consistently, YAP activation was associated with enhanced growth and metastatic potential in vivo [129]. To further confirm the causative role of the carbonyl stress-related compound MGO, all these and other effects, including resistance to epithelial growth factor receptor (EGFR)-targeted therapy in colorectal tumors [161], were reversed by using the RCS trapping agent L-carnosine [129,160,161]. Similar findings on the favorable effect of MGO-mediated protein modifications in cancer progression were previously reported in lung [165] and gastrointestinal [166] cancers and were associated with the formation of MGO-derived AGE structures on heat-shock protein 27 (Hsp27) [165,166]. Again, inhibition of MGO-induced AGE formation on Hsp27 caused sensitization of cancer cells to anticancer drugs [167].

All the above studies have the merit of having identified the molecular mechanisms by which MGO-mediated carbonyl stress may affect cancer growth, therapeutic resistance, and metastasis. However, by considering AGEs as a mere indicator of tumor carbonyl stress, and aerobic glycolysis as the unique source of RCS, these studies overlooked important aspects related to carbonyl stress and, probably, its impact on cancer, particularly in diabetic patients. In fact, in addition to physico-chemical modifications of regulatory proteins mediated by glycolysis-derived RCS in tumor cells, circulating RCS and related AGE structures (i.e., systemic carbonyl stress) might also affect tumor cell behavior, particularly by interacting with RAGE and other receptors on tumor cells [168,169,170,171]. The impact of the overall burden of carbonyl stress in tumor pathology likely involves several RCS other than the glycolytic side-product MGO, including the endogenous RCS derived from both oxidative and non-oxidative metabolism of glucose and lipids (i.e., glyoxal, acrolein and other aldehydes) and RCS/AGE derived from exogenous sources (Figure 2). Therefore, the focus on a single RCS in the absence of a general assessment of the impact of carbonyl stress on cancer limits the understanding and the significance of these interesting data to diabetes and, in general, other carbonyl stress-related conditions that are also widely recognized risk factors for malignancy, such as obesity [44,172], smoking habit [95,96,97], and consumption of high fat and high-temperature-processed food [95,101,173]. This issue may not be purely academic, as several clinical and experimental studies have suggested a role of AGEs in the development and progression of various types of cancers, including breast [174], liver [175,176], colorectal [177], and kidney [178,179]. Most of these and other studies [180] have proposed that the cancer-promoting effect of AGEs is mainly mediated by AGE-RAGE interaction.

## 5. RAGE, AGEs, and Their Carbonyl Precursors as Potential Targets in Pancreatic Cancer Associated with Diabetes and Other Carbonyl Stress-Related Conditions

RAGE was found to be expressed in all cell types implicated in tumor formation [181], including inflammatory cells, and to regulate crosstalk between pro-survival pathways in PDAC cells [182] by interacting with multiple endogenous and exogenous ligands. RAGE has been linked to the development/progression of several cancers by favoring chronic inflammation [183] and promoting tumor growth and metastasis [184]. Consistently, genetic or pharmacologic blockade of RAGE signaling has been demonstrated to suppress carcinogenesis, cancer progression, and spreading [184,185]. Mechanistically, RAGE signaling cooperates with mutant KRAS by activating NF-κB, a critical transcription factor transducing a multitude of inflammatory signals within the cell [186], and sustains mutant KRAS in modulating signaling pathways that control cell survival, proliferation, angiogenesis, and migration [187,188,189,190,191,192,193,194,195,196,197,198]. In turn, NF-κB activation by RAGE signaling and the hypoxic environment upregulates RAGE itself, since the RAGE gene promoter contains functional binding elements for NF-κB [199]. (Figure 5).

There is convincing evidence supporting a critical role for the AGE-RAGE system in human and experimental PDAC (Table 1).

A large prospective study investigated the associations of pre-diagnostic levels of serum CML (a major AGE epitope) and soluble RAGE (sRAGE) with PDAC in a cohort of 29,133 Finnish male smokers. sRAGE is a truncated circulating form lacking the transmembrane and the signaling domain, thus acting as a decoy receptor in preventing RAGE activation and inflammatory signaling cascades [189]. In this study [190], Jiao et al. found that sRAGE levels were inversely associated with the risk of PDAC. Conversely, the CML/sRAGE ratio (i.e., free CML), but not total serum levels of CML (i.e., free CML plus sRAGE-bound CML), were positively associated with PDAC risk. The finding of an inverse relationship between pre-diagnostic sRAGE levels and risk of incident PDAC was recently confirmed in a cohort of postmenopausal women within the prospective Women’s Health Initiative Study [191].

Consistent with human findings, studies conducted in murine models of PDAC have demonstrated a role for RAGE in maintaining oncogenic signaling in PDAC cells by sustaining Kras activity and inflammation. In fact, RAGE inhibition was associated with suppression of NF-kB and extracellular signal-regulated kinases (ERK) activities in cells expressing oncogenic KRAS [192]. Accordingly, deletion or pharmacological blockade of RAGE slows down tumor growth and metastasis [193,194] by delaying noninvasive lesion (i.e., pancreatic intraepithelial neoplasias, PanINs) progression to PDAC [193], thus significantly prolonging survival in these mouse models. Therapeutic targeting of RAGE was shown to exert multiple beneficial effects through a variety of mechanisms, including preventing the accumulation of myeloid-derived suppressor cells in tumor tissue [195], sensitizing PDAC cells to oxidative injury [196], diminishing autophagy and inflammation [197], regulating mitochondrial bioenergetics [198], and modulating the crosstalk between pro-survival pathways in PDAC cells [199]. Together with the demonstration that RAGE protein levels increase in parallel with the progression of PanIN lesions in mice and are higher in human PDAC specimens [193], these findings strongly support RAGE involvement in PDAC growth.

Despite the bulk of evidence indicating a role of RAGE in PDAC, only one study [171] investigated the effects of their natural ligands AGEs on RAGE activity and PDAC development (Table 1). Consistent with a role of the AGE/RAGE axis, exogenous CML administration to a mouse model of Kras-driven PDAC induced RAGE upregulation in PanINs and markedly accelerated progression to PDAC. Compared to coeval vehicle-treated mice, six-week treatment with CML increased the cumulative incidence of PDAC by more than seven times at 11 weeks of age [171]. In vitro mechanistic studies revealed that CML promotes human PDAC cell growth in a concentration-dependent and time-dependent manner by increasing activation of NF-κB and downstream tumorigenic pathways. Importantly, these CML-mediated effects were counteracted by acute RAGE blockade in PDAC cells [171]. Although supporting previous data demonstrating a permissive role for RAGE in early PDAC, and the efficacy of therapeutic targeting of RAGE in delaying PDAC development in naïve mice, these findings argue against the utility of RAGE blockade/inhibition as a therapeutic option in conditions characterized by increased circulating AGE levels, such as diabetes. In fact, RAGE blockade failed to prevent the PDAC-promoting effect of the AGE structure CML in mice. This was associated with PDAC tissue upregulation of the RAGE homologue CD166/activated leukocyte cell adhesion molecule (ALCAM), which shares with RAGE some endogenous ligands [200]. Consistent with this finding, CD166/ALCAM was previously found to be upregulated after genetic deletion of RAGE in the context of tissue inflammation driven by RAGE-ligands [200] and associated with tumor spread and recurrence in human cancers [201,202], including PDAC [202]. Overall, these results support the concept that AGEs promote invasive PDAC through receptor-mediated mechanisms and might be an important mediator of the increased risk of PDAC conferred by diabetes. However, they suggest that an AGE reduction strategy, instead of RAGE inhibition, might be more suitable for the risk management and prevention of PDAC in diabetic patients. A corollary observation is that environmental sources of RCS/AGEs might be also involved in the increased risk of PDAC associated with dietary and smoking habits [95,96,97,99,100,101], including CML derived from red meat, the consumption of which has been linked to an increased risk in men [101].

The hypothesis of inhibition of AGE formation as a possible therapeutic avenue for abating the PDAC risk conferred by diabetes was tested in an experimental model of T1DM [203] (Table 1). After 16 weeks of T1DM (i.e., at 22 weeks of age), the cumulative incidence rate of PDAC was 75%, as compared with 8% in coeval non-diabetic mice [203]. Given that insulin deficiency is the unique etiological factor of T1DM, any effects of diabetes in this experimental model of Kras-driven PDAC could only be ascribed to hyperglycemia. Therefore, from an etiological perspective, the dramatic effect of T1DM on PDAC onset is consistent with the recent epidemiological finding that pancreatic cancer incidence increases linearly with increasing fasting glucose levels, even in populations with normal glucose range [36]. From a mechanistic perspective, and in agreement with previous studies indicating that RCS play a critical role in cancer growth [129,159,160,161,165,166,167], treatment of diabetic mice with the carbonyl trapping agent (and AGE inhibitor) FL-926-16 prevented the accelerating effect of diabetes on PanINs progression to PDAC [203] (Figure 6).

In vitro experiments demonstrated that the PDAC promoting effect of hyperglycemia was mediated by RCS and their irreversible adducts AGEs. Both classes of carbonyl stress-related compounds have proven to be potent inducers of YAP activity [129,203], a key downstream target of KRAS signaling required for progression of PanINs to invasive PDAC [164,204]. However, the intracellular pathways involved in YAP activation by RCS and AGEs were different. In fact, RCS effect was mainly driven by a reduction in Large Tumor Suppressor Kinase 1 levels, which is a negative regulator of YAP activity [205]. Conversely, AGEs exerted their effects on YAP by activating the signaling cascade of EGFR/ERK, which has proven essential for Kras-driven PDAC [206,207].

Overall, these experimental studies provide substantial evidence that carbonyl stress is involved in PDAC development and progression, is responsible for the additional risk conferred by diabetes, and may be a potential pharmacological target in PDAC prevention/treatment, particularly in high-risk diabetic patients.

## 6. Conclusions

There is a consensus in considering long-standing diabetes (onset >36 months before the neoplastic diagnosis) as a risk factor for pancreatic cancer [20,22,25,26,27,29]. However, as discussed above, the highest risk is found within the first two years after diabetes diagnosis, and then it gradually decreases as the duration of diabetes increases [27,28,29]. Given the decreasing association of diabetes with pancreatic cancer risk over time, the controversy regarding the causal role of diabetes has risen, and the theory of reverse causation has been established where PDAC can induce diabetes. While the “reverse causation hypothesis” could explain the diabetes-PDAC relationship in subjects with T2DM, it cannot explain several aspects of the diabetes–cancer relationship, as the same magnitude and temporal trajectory of the risk has been observed in T1DM and, importantly, for cancers other than PDAC, in both T2DM and T1DM.

Given the different etiologies of the two forms of diabetes with respect to insulin availability, the similar association with PDAC and other malignancies also supports the concept that hyperglycemia, rather than hyperinsulinemia, may be the driving force between diabetes and cancer [208]. Besides being biologically plausible, the “hyperglycemic hypothesis” may also explain the decreasing association of diabetes (both T2DM and T1DM) with the risk of PDAC (and other cancers) over time. Indeed, according to this hypothesis, hyperglycemia onset in individuals with pre-existing PanINs—the most common precursors of PDAC—would promote rapid transition to invasive cancer (i.e., PDAC) by increasing carbonyl stress and favoring PanINs progression (Figure 6). Together with the notion that the prevalence of PanINs is about 30% in patients aged 50 years, and further increases with age [209], the hypothesis of hyperglycemia as a causal factor may be consistent with the predominance of PDAC diagnosis in the two-year period following the diagnosis of diabetes (Figure 1). The gradual reduction of cases over the first four-year period would depend on the extent and grade of PanINs at the onset of diabetes, with low grade PanINs taking longer to progress to PDAC compared with high grade PanIN lesions [210]. The residual (but lasting) increased risk of PDAC many years (i.e., >4 years) after the diagnosis of diabetes would be driven by the ability of glucose to act as DNA-damaging factor in PanINs-free subjects at diabetes onset, leading to genomic instability and, eventually, to precancerous lesion formation. About that, the carcinogenic potential of glucose and its role in PDAC initiation have been recently proposed, as high glucose was shown to induce KRAS mutations preferentially in pancreatic cells [211]. As KRAS mutations are virtually present in all PanINs of any grade [210,211], mutant KRAS is considered as a prerequisite for the development of ductal preneoplasia. Experimentally, the hypothesis that diabetes is a causal factor of PDAC and exerts its promoting effect through carbonyl stress, is supported by data demonstrating that hyperglycemia and AGEs dramatically accelerate and increase PDAC development in a mouse model of Kras-driven PaC [171,203] and that treatment of diabetic mice with RCS sequestering agents—and, thus, AGE inhibitors—prevents PanIN progression to PDAC induced by diabetes [203].

Based on the currently available experimental data, L-carnosine and its derivatives have proven effective in countering cancer progression, aggressiveness, and therapeutic resistance [129,160,161,167] and in preventing the PDAC-promoting effect of diabetes [203]. Altogether, these findings support the concept that carbonyl stress is critical in cancer development and growth and represents the mechanistic driver between diabetes and PDAC, mainly by favoring PanIN progression. However, as PDAC results from the gradual accumulation of genetic alterations and it may take many years to transform cells into invasive/metastatic cancer [212], the role of diabetes- and prediabetes-associated carbonyl stress in cancer initiation should be elucidated in future studies. Finally, as carbonyl stress is a harmful condition associated with several metabolic disorders [81,111], and can be fueled by bad habits like smoking and consumption of ultra-processed food, its role as common mechanistic link and the benefit of anti-RCS/AGE drugs in treating and preventing PDAC deserve further investigation in different experimental models of carbonyl stress-related conditions, in addition to diabetes.

## Figures and Tables

**Figure 1 cancers-13-00313-f001:**
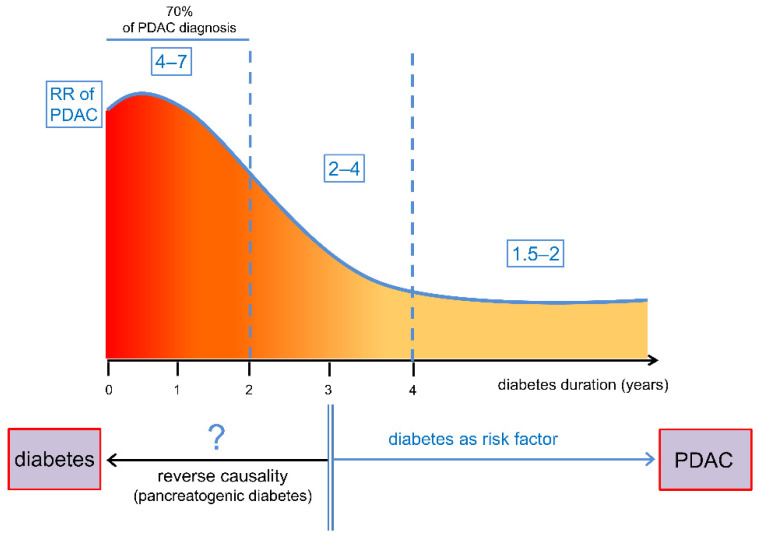
Relationship between diabetes and pancreatic cancer. The epidemiological association between diabetes and pancreatic ductal adenocarcinoma (PDAC) is complex, leading to divergent and even opposing views on the nature of the relationship. Please refer to the main text for details and references. RR = relative risk.

**Figure 2 cancers-13-00313-f002:**
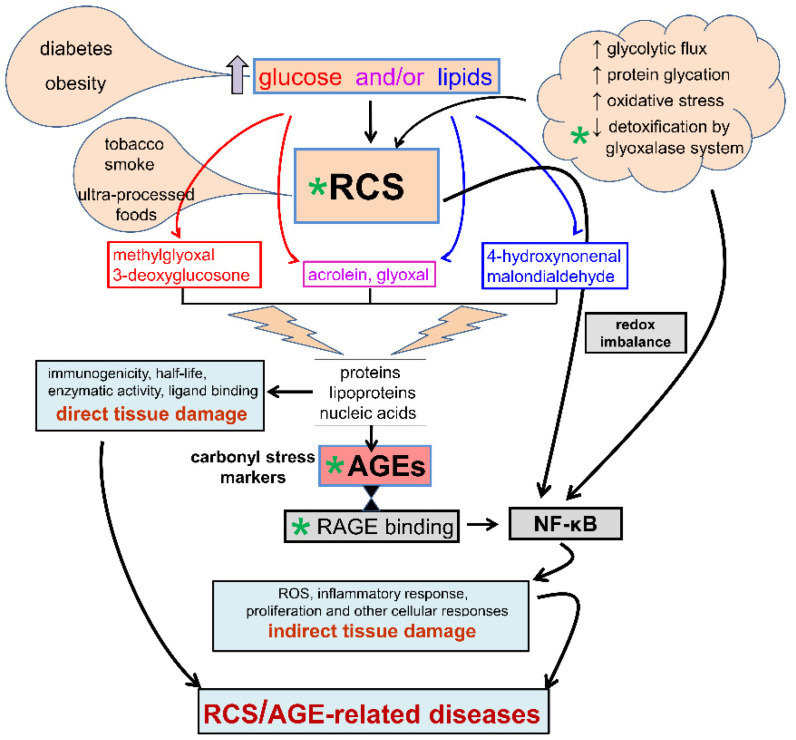
Carbonyl stress. Causes and mechanisms of reactive carbonyl species (RCS) and advanced glycation end-products (AGEs) formation, pathogenetic mechanisms in carbonyl stress-induced tissue damage, and potential drug targets (green asterisk). NF-κB = nuclear factor-κB; ROS = reactive oxygen species.

**Figure 3 cancers-13-00313-f003:**
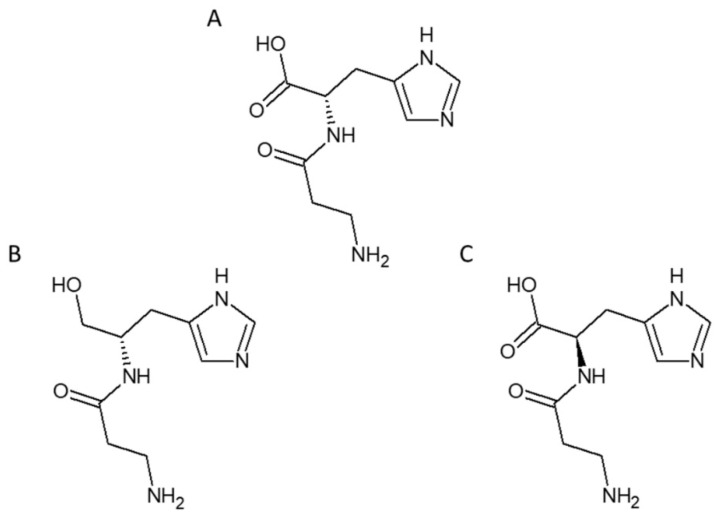
Carnosines with RCS scavenging activity. Chemical structure of L-carnosine (β-alanyl-L-histidine) (**A**), the carnosinase-resistant carnosine derivative carnosinol (FL-926-16; (2S)-2-(3-amino propanoylamino)-3-(1H-imidazol-5-yl) propanol) (**B**), and the enantiomer D-carnosine (β-alanyl-D-histidine) (**C**).

**Figure 4 cancers-13-00313-f004:**
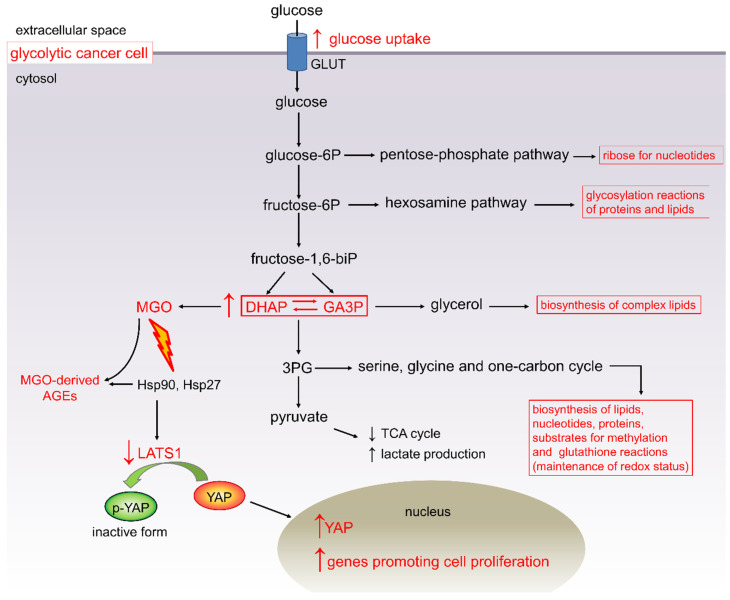
Aerobic glycolysis (Warburg effect) and cell proliferation. In addition to supporting cell growth by feeding several non-mitochondrial anabolic pathways, enhanced glucose flux through the glycolytic pathway directly stimulates cell proliferation through overproduction of the reactive carbonyl species (RCS) methylglyoxal (MGO), which is an inevitable by-product of glycolysis. GLUT = glucose transporter; DHAP = dihydroxyacetone phosphate; GA3P = glyceraldehyde 3-phosphate; 3PG = 3-phosphoglycerate; AGEs = advanced glycation end-products; TCA = tricarboxylic acid; LATS1 = large tumor suppressor kinase 1; YAP = yes-associated protein.

**Figure 5 cancers-13-00313-f005:**
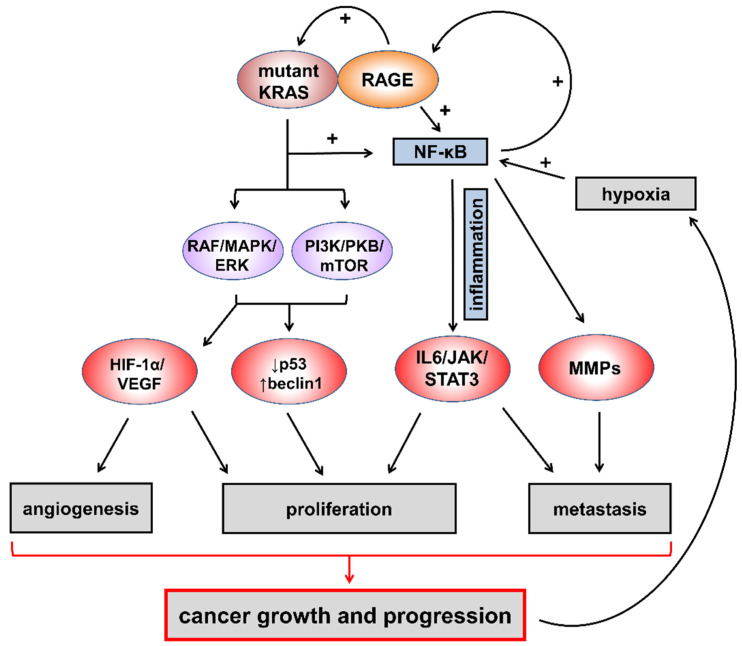
RAGE signaling cooperates with mutant KRAS in pancreatic cancer development and progression. RAGE signaling results in upregulation of several KRAS-dependent pathways playing a critical role in PDAC development and progression. RAGE = receptor for advanced glycation end-products; NF-κB = nuclear factor-κB; RAF/MAPK/ERK = rapidly accelerated fibrosarcoma/mitogen-activated protein kinase/extracellular signal-regulated kinases; PI3K/PKB/mTOR = phosphoinositide 3-kinase/protein kinase B/mammalian target of rapamycin; HIF-1α/VEGF = hypoxia-inducible factor 1α/vascular endothelial growth factor; p53 = tumor protein P53; IL6/JAK/STAT3 = interleukin 6/janus kinase/signal transducer and activator of transcription 3; MMPs = matrix metalloproteinases.

**Figure 6 cancers-13-00313-f006:**
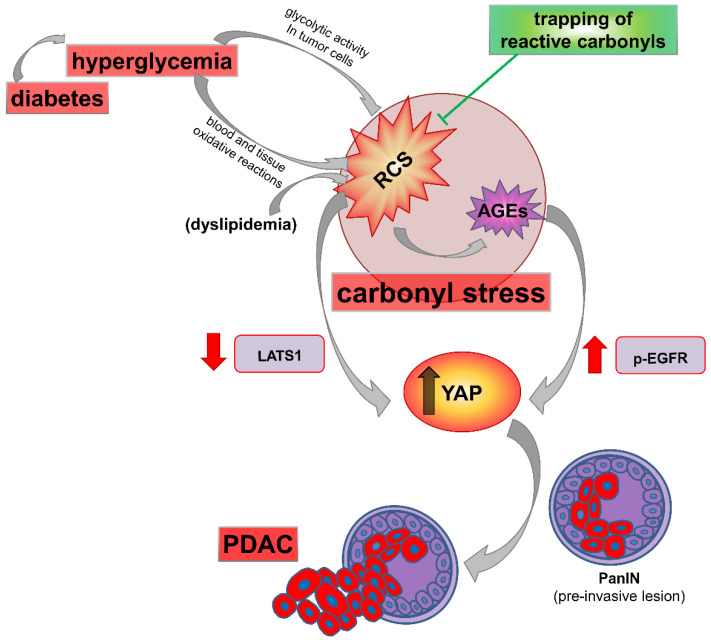
Carbonyl stress in pancreatic ductal adenocarcinoma (PDAC) promotion induced by diabetes, and mechanism of protection by reactive carbonyl species (RCS) trapping agents. Diabetes markedly accelerates tumor progression through hyperglycemia-derived carbonyl stress. The increased availability of glucose feeds the glycolytic flux of tumor cells favoring local formation of RCS such as methylglyoxal, which represent an inevitable side-product of glycolysis. In addition, circulating RCS and advanced glycation end-products (AGEs) derived from non-enzymatic glycoxidation (and lipoxidation) reactions occurring at the systemic level may also contribute to the overall burden of carbonyl stress in neoplastic lesions. Increased RCS and AGE levels in PDAC cells has been associated with increased nuclear translocation of Yes-associated protein (YAP), a key effector of Hippo pathway and regulator of tumor growth and invasion. Dyslipidemia, associated or not with hyperglycemia, may represent an additional source of RCS and AGEs. Sequestering of RCS and consequent inhibition of AGE formation efficiently prevented hyperglycemia-induced YAP activation and acceleration of pancreatic intraepithelial neoplasia (PanIN) progression to invasive cancer in a mouse model of Kras-driven PDAC, through inhibition of large tumor suppressor kinase 1 (LATS1) and phosphorylation of epidermal growth factor receptor (p-EGFR), respectively.

**Table 1 cancers-13-00313-t001:** Clinical and experimental studies on potential drug targets related to carbonyl stress in pancreatic ductal adenocarcinoma (PDAC).

Study	Design/Intervention	Population/Animal or Cell Culture Model	Drug Target	Main Result
Clinical				
Jiao et al. [190]	prospective case-cohort study	Finnish male smokers	AGE/RAGE axis	pre-diagnostic sRAGE is inversely associate while CML AGE:sRAGE ratio is positively associated with PDAC risk
White et al. [191]	prospective nested case-control study	postmenopausal women	AGE/RAGE axis	pre-diagnostic sRAGE is inversely associated with pancreatic cancer risk
Kahlert et al. [202]	retrospective study	PDAC patients undergoing potentially curative resection	ALCAM/CD166 (RAGE homolog)	ALCAM/CD166 is an independent prognostic marker for survival and tumour relapse in PDAC
Jiao et al. [101]	prospective study	NIH-AARP Diet and Health Study participants	AGEs	dietary AGE consumption is associated with increased risk of PDAC
Experimental				
Kang et al. [193]	RAGE knock-down	murine model of Kras-driven PDAC and human PDAC tissue	RAGE-IL6-pSTAT3 pathway	RAGE ablation delays PDAC development by decreasing STAT3 signaling and autophagy
Arumugam J et al. [194]	RAGE blockade	pancreatic orthotopic model	RAGE-NF-κB axis	RAGE inhibition reduces PDAC growth and metastasis
Azizan et al. [192]	RAGE blockade	pancreatic orthotopic model	RAGE-NF-κB-KRAS axis	RAGE inhibition lowers oncogenic KRAS activity by preventing NF-κB activation
Vernon et al. [195]	RAGE knock-down	murine model of Kras-driven PDAC	RAGE-IL6 pathway	RAGE ablation delays PDAC development by reducing the accumulation of myeloid-derived suppressor cells
Kang et al. [196]	RAGE knock-down or inhibition by RNA interference	PDAC cells	ROS-NF-κB-RAGE axis	suppression and knockdown of RAGE increases the sensitivity of PDAC cells to oxidative injury
Kang et al. [197]	RAGE knock-down	murine model of Kras-driven PDAC	RAGE-IL6-pSTAT3 pathway	RAGE ablation increases apoptosis and decreases autophagy/proliferation in the emerging PDAC microenvironment
Kang et al. [198]	RAGE knock-down or inhibition of HMGB1 release	ectopic tumor xenograft model and human PDAC tissue	HMGB1–RAGE axis	lack of RAGE or inhibition of HMGB1 release slows PDAC growth in vitro and in vivo by diminishing ATP production
Kang et al. [182]	RAGE knock-down	murine model of Kras-driven PDAC	RAGE-NF-κB-KRAS pathway	binding of RAGE to oncogenic KRAS facilitates HIF-1α activation and promotes PDAC growth
Menini et al. [171]	exogenous AGE administration/RAGE blockade	murine model of Kras-driven PDAC	AGE-RAGE-ALCAM/CD166 axis	AGEs accelerate the progression of PDAC through receptor-mediated mechanisms
Menini et al. [203]	inhibition of AGE formation by RCS scavenging	diabetic murine model of Kras-driven PDAC and human PDAC tissue	RCS (AGE precursors)	circulating and tumor-derived RCS/AGEs generated by hyperglycemia promote invasive PDAC

AGE = advanced glycation end-products; RAGE = Receptor for RAGE; sRAGE = soluble RAGE; ALCAM/CD166 = activated leukocyte cell adhesion molecule/cluster of differentiation 166; IL6 = interleukin 6; STAT3 = phosphorylated signal transducer and activator of transcription 3; NF-κB = nuclear factor-κB; ROS = reactive oxygen species; HMGB1 = High Mobility Group Box 1; ATP = adenosine triphosphate; HIF-1α = Hypoxia-inducible factor 1-α; RCS = reactive carbonyl species.

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
