# Peer review of "Diabetes and Pancreatic Cancer—A Dangerous Liaison Relying on Carbonyl Stress"

_cancers, 2021, doi:10.3390/cancers13020313_

Round 1
Reviewer 1 Report
Dear Authors
I have the following observations:
- section 4 (carbonyl stress in cancer: a possible link between metabolism and malignances), although important for the understanding of the subsequent sections of the paper, deals mainly with cancer in general (and does not focus on pancreatic cancer) and could be shortened
- it would be interesting to add a table which summarizes the current preclinical/clinical knowledge on potential drug target
-there are some typos (e.g. in favor-of it line119; assumption- line134, mediates-->mediate line167)
Kind Regards
Author Response
Section 4 (carbonyl stress in cancer: a possible link between metabolism and malignances), although important for the understanding of the subsequent sections of the paper, deals mainly with cancer in general (and does not focus on pancreatic cancer) and could be shortened.
As requested by the Reviewer, Section 4 (“Carbonyl stress in cancer: a possible link between metabolism and malignances”) has been shortened by eliminating redundant information on carbonyl stress and cancer in general. We have also included a short paragraph on glucose metabolism in PDAC cells, as requested by Reviewer #2.
It would be interesting to add a table which summarizes the current preclinical/clinical knowledge on potential drug target.
As suggested by the Reviewer, we have added a table summarizing the current clinical/preclinical studies on potential drug targets related to carbonyl stress in pancreatic cancer.
There are some typos (e.g., in favor of it - line119; assumption - line134, mediates-->mediate - line167).
Thanks for indicating these typos. We have now corrected these and other typos throughout the manuscript.
Reviewer 2 Report
Menini and co-workers provide a review manuscript entitled “Diabetes and pancreatic cancer - a dangerous liaison relying on carbonyl stress”. Based on epidemiological reports having revealed a link between type 2 (T2DM) as well as type 1 (T1DM) diabetes mellitus and risk of pancreatic cancer the authors pinpoint key drivers involved in the cancerous process. The authors focus on carbonyl stressors in relation to the dysmetabolic state seen in diabetic subjects as potential mechanism in relation to onset and progression of pancreatic cancer. Finally, potential therapeutic avenues in relation to carbonyl stressors are discussed such as mitigation of oncogenic pathways using carbonyl-scavenging agents and AGE inhibitors. In summary, the authors provide an interesting view on a highly fatal malignancy, which reveals increasing age-standardized incidence rates worldwide and which is unfortunately characterized by poor survival rates.
Specific comments:
- This is an interesting and well-balanced review covering the link between DM and pancreatic cancer.
- In section “3. Diabetes, carbonyl stress, advanced glycation end-products (AGEs) formation and related therapeutic strategies” the authors should be more specific and address for example the potential impact of a dysregulated NF-kB pathway in relation to carbonyl stress.
- “Figure 2. Carbonyl stress” should consequently indicate the specific key pathways involved, such as the RAGE-NF-kappaB pathway activation in response to oxidative stressors. In addition, the role of NF-kappaB/Rel transcription factors in pancreatic cancer as well as pancreatic cancer metastasis should be covered in Figure 2. Also in section 3 of the draft this important pathway should be covered in more detail.
- Section 4. “Carbonyl stress in cancer: a possible link between metabolism and malignancies” would benefit from sharing information on in vivo glucose metabolism of pancreatic cancer cells, namely the remarkable sensitivity of FDG PET/CT scanning of pancreatic malignancies indicative of high glucose utilization by the cancerous cells. Also, the review paper by Yan L et al. (Cancers 2019;11:1460) covering in detail glucose metabolism in pancreatic cancer should be cited in this context.
- General remark: The authors largely focus on endogenous sources as modulators of carbonyl stress and the formation of advanced glycation end products. Exogeneous sources of carbonyl stress are not covered in the review. Therefore, the major source of these products, such as the exogeneous uptake of carbonyl stress components (for example smoking, ingestion of highly processed, AGE rich food etc.), should also be mentioned. Of note, this issue is quite comprehensively covered by a review in Seminars in Cancer Biology (Saheem Ahmad et al. “DOI:10.1016/j.semcancer.2017.10.012 – and 2018;49:44).
Author Response
Menini and co-workers provide a review manuscript entitled “Diabetes and pancreatic cancer - a dangerous liaison relying on carbonyl stress”. Based on epidemiological reports having revealed a link between type 2 (T2DM) as well as type 1 (T1DM) diabetes mellitus and risk of pancreatic cancer the authors pinpoint key drivers involved in the cancerous process. The authors focus on carbonyl stressors in relation to the dysmetabolic state seen in diabetic subjects as potential mechanism in relation to onset and progression of pancreatic cancer. Finally, potential therapeutic avenues in relation to carbonyl stressors are discussed such as mitigation of oncogenic pathways using carbonyl-scavenging agents and AGE inhibitors. In summary, the authors provide an interesting view on a highly fatal malignancy, which reveals increasing age-standardized incidence rates worldwide and which is unfortunately characterized by poor survival rates. This is an interesting and well-balanced review covering the link between DM and pancreatic cancer.
We thank the Reviewer for providing us with positive comments and useful suggestions.
In section “3. Diabetes, carbonyl stress, advanced glycation end-products (AGEs) formation and related therapeutic strategies” the authors should be more specific and address for example the potential impact of a dysregulated NF-kB pathway in relation to carbonyl stress.
This important point concerning NF-kB pathway dysregulation by carbonyl stress has been addressed in the revised version of the manuscript (see lines 265-269 and modified Figure 2).
“Figure 2. Carbonyl stress” should consequently indicate the specific key pathways involved, such as the RAGE-NF-kappaB pathway activation in response to oxidative stressors. In addition, the role of NF-kappaB/Rel transcription factors in pancreatic cancer as well as pancreatic cancer metastasis should be covered in Figure 2. Also in section 3 of the draft this important pathway should be covered in more detail.
This topic, which had already been addressed in the original version of the manuscript (see lanes 479-490), has been expanded in the revised version (see lines 265-269, 424-431, and 460-463, modified Figure 2 and new Figure 5). However, we preferred to address the role of NF-kB and other pathways activated by RAGE in pancreatic cancer (see also our response to Reviewer #3) in a new figure (Figure 5 of the revised version) and in Section 5, which deals with the role of carbonyl stress in pancreatic cancer. In this way, Section 3 and Figure 2 remain dedicated to introducing diabetes as a source of carbonyl stress, whereas Section 5 describes the role of carbonyl stress in PDAC, as originally conceived.
Section 4. “Carbonyl stress in cancer: a possible link between metabolism and malignancies” would benefit from sharing information on in vivo glucose metabolism of pancreatic cancer cells, namely the remarkable sensitivity of FDG PET/CT scanning of pancreatic malignancies indicative of high glucose utilization by the cancerous cells. Also, the review paper by Yan L et al. (Cancers 2019;11:1460) covering in detail glucose metabolism in pancreatic cancer should be cited in this context.
Thanks for this suggestion. This relevant issue concerning increased glucose metabolism in PDAC cells has been discussed (see lanes 334-338) and the manuscript by Yan L et al. has been cited (see Ref #148) in the revised manuscript.
General remark: The authors largely focus on endogenous sources as modulators of carbonyl stress and the formation of advanced glycation end-products. Exogeneous sources of carbonyl stress are not covered in the review. Therefore, the major source of these products, such as the exogeneous uptake of carbonyl stress components (for example smoking, ingestion of highly processed, AGE rich food etc.), should also be mentioned. Of note, this issue is quite comprehensively covered by a review in Seminars in Cancer Biology (Saheem Ahmad et al. “DOI:10.1016/j.semcancer.2017.10.012 – and 2018;49:44).
This issue has been addressed in the revised version of the manuscript (see lines 250-253, 422, and 496-499), in which we have extended and elaborated on the information already present in the original version (see lanes 410-413 and Figure 2). The articles suggested have been cited (see Refs #99 and 100). Please, see also our response to Reviewer #4.
Reviewer 3 Report
Overall, this is an interesting well written review with important clinical relevance. However, I have several concerns that need to be addressed:
- As most of the PDAC present with KRAS mutations, the effect of RCS and AGEs should be cooperate with KRAS signaling, which was not presented or discussed from the angles, either T1DM/T2DM as risk factors for PDAC, or T1DM and T2DM actually promote for PDAC.
- The RAGE signaling pathway was not well discussed under the AGEs and KRAS WT or mutation scenario.
- I understand that the authors want to focus more on the LATS1-YAP signaling pathway in the PDAC context. However the AGEs-RAGE signaling pathway should utilize multiple downstream signaling pathways in the pancreatic cells. Therefore, an overview of the key signaling pathways may give readers a better and unbiased view of the potential impacts on PDAC development.
Author Response
Overall, this is an interesting well written review with important clinical relevance. However, I have several concerns that need to be addressed:
We thank the Reviewer for her/his overall positive judgement and valuable suggestions and feedback in improving our manuscript.
As most of the PDAC present with KRAS mutations, the effect of RCS and AGEs should cooperate with KRAS signaling, which was not presented or discussed from the angles, either T1DM/T2DM as risk factors for PDAC, or T1DM and T2DM actually promote for PDAC.
Actually, we already discussed the effects of RCS and AGEs on YAP and EGFR, which are key downstream targets of KRAS signaling (see lanes 386-389 and 528-535). In the revised version of the manuscript, we have expanded the discussion of the role of RCS, AGEs and RAGE in sustaining KRAS activity through NF-κB activation (see lanes 265-269, 426-431, and 461-463, and new Figure 5).
The concept that diabetes in general (both T1DM and T2DM) promotes PDAC by favoring rapid transition of PanINs to invasive cancer is discussed in the “Conclusions” section (see lines 554-562 and Figure 6) of the revised version. This is the mechanism that we propose to explain the increased risk of PDAC diagnosis in diabetic patients, particularly in the two-year period following the diagnosis of diabetes (i.e., diabetes as a risk factor for PDAC and not vice versa). Moreover, the promoting vs the carcinogenic effect of hyperglycemia (i.e., diabetes) is also discussed in lines 564-569 of the revised version, including the ability of hyperglycemia to induce KRAS mutations.
The RAGE signaling pathway was not well discussed under the AGEs and KRAS WT or mutation scenario. I understand that the authors want to focus more on the LATS1-YAP signaling pathway in the PDAC context. However, the AGEs-RAGE signaling pathway should utilize multiple downstream signaling pathways in the pancreatic cells. Therefore, an overview of the key signaling pathways may give readers a better and unbiased view of the potential impacts on PDAC development.
This relevant issue has been addressed in the revised version of the manuscript (see lanes 265-269, 420-431, and 461-463, modified Figure 2 and new Figure 5), in which we have extended and elaborated on the information already present in the original version (see lanes 479-483 and 528-535). We have addressed the role of NF-kB and other pathways activated by RAGE (see also our response to Reviewer #2) in pancreatic cancer in a new figure (see Figure 5 of the revised version) and in Section 5, which deals with the role of carbonyl stress (i.e., RAGE, AGEs and RCS) in pancreatic cancer.
Reviewer 4 Report
The manuscript submitted by Menini et al. nicely compile the recent findings on the relationship between diabetes mellitus and the development of PDAC. The manuscript is well written and well organized. It starts introducing the details on the interplay between T1DM and T2DM with PDAC and prove that both have a common factor that seems to trigger PDAC, that is hyperglycemia. Then it discusses how obesity might also play an important role within this game, but also how diabetic treatments might unbalance the interrelationship. Furthermore, it discusses how RCS (specially MGO), appearing as side products on an enhanced glycolysis, become the triggering factors of PDAC as a result of RAGE activation. This reviewer find that the information provided in this review is relevant for those investigations focused in try to better understand the relation between diabetes and PDAC. Consequently, I would recommend its acceptance for publication in Cancers. However, before the final acceptance, I would ask the authors to reply to some questions that came to my mind while I was reading the manuscript, and modify the manuscript accordingly.
1.- The authors focused their review on the effect of diabetes on PDAC. I understand their point since PDAC represents the 90% of pancreatic cancers. However, this reviewer was expecting to have some insights on the relationship between diabetes and the other type of pancreatic cancers. Even if there is not yet a clear correlation I think the authors should briefly comment on that in their manuscript.
2.-This reviewer was surprised that the higher risk for diabetic people to develop PDAC is within the first two years after diabetes was diagnosed. This reviewer (and likely most of the audience) was curious to know why this happens. However, I could not find a clear explanation about that in the manuscript. I think it would be ideal if the authors could provide mechanistic information on that. In case this is not available, I think it will be good if they could raise a hypothesis.
3.-The authors state that a possible therapeutic strategy against the development of diabetes-related PDAC might be the use of compounds capable to scavenge RCS. Although there are many compounds able to reduce the concentration of RCS, they just focused on L-carnosine saying that it has been the most studied compound since it does not have safety issues. This reviewer understand their point, but also find mandatory to expand this section including other RCS scavengers and their possible effect on PDAC related to diabetes. For instance one of the most powerful AGE inhibitor is pyridoxamine. The authors should comment on that or on others AGEs inhibitors and their potential on PDAC.
4.-The authors should include an additional figure showing the chemical formula of L-carnosine and that of F-926-16. This will help the reader to better understand the structural differences between them.
5.-This reviewer also find interesting that the authors slightly expand their manuscript responding the effects of exogenous AGEs on the development of pancreatic cancer. As example, they could comment on the results published in this manuscript: Am J Clin Nutr 2015 Jan;101(1):126-34. doi: 10.3945/ajcn.114.098061, but also on the results provided in others. If they did not do that due length limitations, I highly recommend to do it but also to remove redundant information appearing in the text.
Author Response
The manuscript submitted by Menini et al. nicely compile the recent findings on the relationship between diabetes mellitus and the development of PDAC. The manuscript is well written and well organized. It starts introducing the details on the interplay between T1DM and T2DM with PDAC and prove that both have a common factor that seems to trigger PDAC, that is hyperglycemia. Then it discusses how obesity might also play an important role within this game, but also how diabetic treatments might unbalance the interrelationship. Furthermore, it discusses how RCS (specially MGO), appearing as side products on an enhanced glycolysis, become the triggering factors of PDAC as a result of RAGE activation. This reviewer find that the information provided in this review is relevant for those investigations focused in try to better understand the relation between diabetes and PDAC. Consequently, I would recommend its acceptance for publication in Cancers. However, before the final acceptance, I would ask the authors to reply to some questions that came to my mind while I was reading the manuscript, and modify the manuscript accordingly.
We thank the Reviewer for her/his overall positive judgement and valuable suggestions and feedback in improving our manuscript.
1. The authors focused their review on the effect of diabetes on PDAC. I understand their point since PDAC represents the 90% of pancreatic cancers. However, this reviewer was expecting to have some insights on the relationship between diabetes and the other type of pancreatic cancers. Even if there is not yet a clear correlation, I think the authors should briefly comment on that in their manuscript.
The reason why we focused on PDAC is because this review article is part of a special issue on PDAC entitled: “Recent Advances in Pancreatic Ductal Adenocarcinoma”. However, we have now briefly mentioned the association of diabetes with other types of pancreatic cancer (see lanes 92-96)
2. This reviewer was surprised that the higher risk for diabetic people to develop PDAC is within the first two years after diabetes was diagnosed. This reviewer (and likely most of the audience) was curious to know why this happens. However, I could not find a clear explanation about that in the manuscript. I think it would be ideal if the authors could provide mechanistic information on that. In case this is not available, I think it will be good if they could raise a hypothesis.
The current prevailing hypothesis to explain the higher risk for PDAC in the first two years after diabetes diagnosis is that early (undiagnosed) PDAC causes diabetes (“reverse causation” hypothesis, see Refs #6,13,15,24, and 30), which we critically discussed in Section 2 (“Relationship between diabetes and pancreatic cancer”) and at the beginning of Section 6 (“Conclusions”). In Section 6, we also present our alternative hypothesis (see lanes 556-569), based on recent data showing that diabetes promotes invasive pancreatic cancer by increasing carbonyl stress in a mouse model of Kras-driven PDAC and that glucose can induce KRAS mutations preferentially in pancreatic cells. In summary, according to this hypothesis (“hyperglycemic hypothesis”), hyperglycemia onset in individuals with pre-existing PanINs ‒ which are about 30% of the general population at age fifty, a percentage that increases further in older subjects ‒ and the consequent increase in carbonyl stress would promote rapid transition to invasive cancer (i.e., PDAC) by favoring PanINs progression.
3. The authors state that a possible therapeutic strategy against the development of diabetes-related PDAC might be the use of compounds capable to scavenge RCS. Although there are many compounds able to reduce the concentration of RCS, they just focused on L-carnosine saying that it has been the most studied compound since it does not have safety issues. This reviewer understand their point, but also find mandatory to expand this section including other RCS scavengers and their possible effect on PDAC related to diabetes. For instance, one of the most powerful AGE inhibitor is pyridoxamine. The authors should comment on that or on others AGEs inhibitors and their potential on PDAC.
According to the Reviewer’s suggestion, we have addressed this point in the revised version of the manuscript (see lines 279-288).
4. The authors should include an additional figure showing the chemical formula of L-carnosine and that of F-926-16. This will help the reader to better understand the structural differences between them.
We have now included an additional figure showing the chemical formula of L-carnosine, FL-926-16, and the enantiomer D-carnosine (see new Figure 3 of the revised manuscript).
5. This reviewer also find interesting that the authors slightly expand their manuscript responding the effects of exogenous AGEs on the development of pancreatic cancer. As example, they could comment on the results published in this manuscript: Am J Clin Nutr 2015 Jan;101(1):126-34. doi: 10.3945/ajcn.114.098061, but also on the results provided in others. If they did not do that due length limitations, I highly recommend to do it but also to remove redundant information appearing in the text.
This issue has been addressed in the revised version of the manuscript (see lines 250-253, 422, and 496-499), in which we have extended and elaborated on the information already present in the original version (see lanes 405-408 and Figure 2) and cited the article suggested by the Reviewer (see Ref #101). Please, see also our response to Reviewer #2.
Reviewer 5 Report
This is a comprehensive and well-written review paper on this topic. However, some minor concerns should be addressed:
(i) Figure 1, the wording of “negative correlation” may be misleading and could suggest to be re-phrased to “the trend of positive correlation is decreased”. How about PPAR inhibitors’ roles in this figure? Some diabetes treatment drugs have been implicated to be able to prevent cancers, for example the PPAR-gamma inhibitors (glitazones) for T2DM, therefor, the diabetes status and the use of glitazones have contradicting effects on cancer formation, the decreased trend of positive correlation may possibly explained, at least in part, by these confounding interactions. It is suggested that the authors elaborate on this point. (ii) Why T3c diabetes is usually mixed with T2DM but not T1DM? It appears that both T1DM and T3cDM, but not T2DM, have problems making insulin. (iii) Fig4, please correct the typo of Carbonyl.
Author Response
This is a comprehensive and well-written review paper on this topic. However, some minor concerns should be addressed:
We thank the Reviewer for providing us with positive comments and useful suggestions.
Figure 1, the wording of “negative correlation” may be misleading and could suggest to be re-phrased to “the trend of positive correlation is decreased”. How about PPAR inhibitors’ roles in this figure? Some diabetes treatment drugs have been implicated to be able to prevent cancers, for example the PPAR-gamma inhibitors (glitazones) for T2DM, therefore, the diabetes status and the use of glitazones have contradicting effects on cancer formation, the decreased trend of positive correlation may possibly be explained, at least in part, by these confounding interactions. It is suggested that the authors elaborate on this point.
“Negative correlation” was referred to “diabetes duration”. However, we have now changed the sentence to “decreasing association of diabetes with pancreatic cancer risk over time”.
Regarding diabetes medications, we have now expanded the paragraph on the effects of these drugs (including thiazolidinediones) on pancreatic cancer risk associated with diabetes (lanes 193-200). Apart from the conflicting evidence concerning their real effects, the use of thiazolidinediones cannot help explain in any way the decreasing association of T2DM and T1DM with pancreatic cancer risk over time, as these drugs are indicated for the treatment of DMT2, but not of T1DM.
Why T3c diabetes is usually mixed with T2DM but not T1DM? It appears that both T1DM and T3cDM, but not T2DM, have problems making insulin.
Thank you for this comment. According to the current classification of diabetes mellitus, type 1 diabetes (T1DM), type 2 diabetes (T2DM), and pancreatic diabetes (type 3c diabetes) are defined as follows (see American Diabetes Association. Classification and Diagnosis of Diabetes: Standards of Medical Care in Diabetes – 2021. Diabetes Care. 2021;44(Suppl. 1):S15–S33):
- T1DM: autoimmune β-cell destruction, usually leading to absolute insulin deficiency;
- T2DM: progressive loss of adequate β-cell insulin secretion on the background of insulin resistance;
- Type 3c diabetes: both structural and functional loss of glucose-normalizing insulin secretion (associated with loss of glucagon secretion) in the context of exocrine pancreatic dysfunction, commonly misdiagnosed as T2DM.
Thus, all these 3 forms of diabetes, including T2DM, have problems to make insulin. In particular, in T2DM, hyperglycemia develops only when pancreatic β-cells are no longer able to secrete enough insulin to compensate for the reduced peripheral action due to insulin resistance. Type 3c diabetes is the consequence of several etiologies, including pancreatitis (acute and chronic), trauma, pancreatectomy, neoplasia, cystic fibrosis, hemochromatosis, fibrocalculous pancreatopathy, rare genetic disorders, and idiopathic forms. In most of these conditions, including neoplasia, there is only a partial loss of β-cells and insulin secretory capacity, just as in T2DM, and development of hyperglycemia depends on the balance between insulin secretion and sensitivity; this is why type 3c diabetes and also latent autoimmune diabetes of adulthood are often misdiagnosed as T2DM. Only in a few instances, e.g., after total pancreatectomy, residual β-cell mass and function is minimal and insufficient per se to maintain glucose homeostasis, thus making type 3c diabetes more similar to T1DM.
Fig4, please correct the typo of Carbonyl.
The typo in Figure 4 (Figure 6 of the revised manuscript) has now been corrected.
Round 2
Reviewer 2 Report
Questions raised have been answered in full.
Reviewer 3 Report
It is a nice review.